# Cost-effectiveness of the ONCORAL multidisciplinary programme for the management of outpatients taking oral anticancer agents at risk of drug-related event: protocol for a pragmatic randomised controlled study

Laure Huot [1,2] Pascale Guerre,[1,3] Guillaume Descotes,[4] Anne-Gaëlle Caffin,[4] Chloé Herledan,[4,5] Florence Ranchon,[4,5] Catherine Rioufol[4,5]

FR and CR contributed equally.

For numbered affiliations see end of article.

**Correspondence to**
Catherine Rioufol;
catherine.rioufol@chu-lyon.fr

## ABSTRACT

**Introduction** The development of oral anticancer agents (OAA) has profoundly changed cancer care, leading patients to manage their chemotherapy treatment on an outpatient basis. The prevention of iatrogenic effects of OAA remains a major concern, especially since their side effects are not less serious than those of intravenous chemotherapy. The ONCORAL programme was set up to secure the management of OAA in cancer patients followed at the Lyon University Hospital. This multidisciplinary programme involves hospital pharmacists, nurses, oncologists, and haematologists, as well as community health professionals. Given the economic stakes that this programme entails for the health system, a medico-economic study was designed.

**Methods and analysis** This is a prospective controlled study, with individual open-label randomisation. A total of 216 outpatients treated with OAA and at risk of developing a drug-related iatrogenic event, will be randomised (2:1) to undergo follow-up in the ONCORAL programme or usual care. The primary outcome will be the estimation of the incremental cost-effectiveness ratio (difference in total costs per quality adjusted life years gained) at 12 months between the two groups. The secondary outcomes will be evaluation of OAA management consequences (relative-dose intensity, adherence, adverse drug events, drug–drug interactions, and proven medication errors), evaluation of overall survival and cancer-related quality of life, and patient-reported outcomes in relation to the treatment. A budget impact analysis will be implemented. Patient and health professional satisfaction regarding the ONCORAL programme will be measured.

**Ethics and dissemination** Approval to conduct this study was obtained from an Ethics Committee (*Comité de Protection des Personnes Ile-de-France VI*) in October 2019, and from the French data protection agency (*Commission Nationale de l'Informatique et des Libertés*), according to the French Law. Trial results will be disseminated at clinical conferences and published in peer-reviewed journals.

**Trial registration** NCT03660670.

## STRENGTHS AND LIMITATIONS OF THIS STUDY

⇒ This study is the first cost-effectiveness evaluation of a multidisciplinary medication management programme for adult cancer outpatients treated with oral anticancer agents.
⇒ A pragmatic open-label randomised design was adopted, based on a programme implemented in a French University Hospital, involving hospital health professionals (oncologists, haematologists, pharmacists, and nurses) and based on a structured relationship with the community health professionals who follow the patient.
⇒ The cost-effectiveness will be estimated at 12 months, by comparing the difference in total costs and quality of life-adjusted years gained between the ONCORAL programme and usual care.
⇒ Due to the nature of the interventions, patients and professionals participating in the study are not blinded to management allocation.

## INTRODUCTION

For many years, the treatment of cancer patients remained essentially hospital-based, relying on surgery, radiotherapy, and injectable chemotherapy. Since the early 2000s, oral anticancer agents (OAA), mostly targeted therapies such as tyrosine kinase inhibitors, have been steadily developed, contributing to significantly improve the prognosis of cancer patients.[1]

By allowing patients previously treated by injectable chemotherapy to receive oral therapy, the use of OAA has shifted patient management from hospital and healthcare provider administered treatment to self-administered treatment at home.[2] Despite their high price, OAA were expected to reduce certain healthcare costs and were

considered for their convenience and ease of administration.[3 4]

However, these drugs expose patients to iatrogenic risks, in part due to their narrow therapeutic index and complex pharmacological profile.[5] They may also result in drug-related problems such as medication errors, adverse effects, and drug–drug interactions, which have been reported in half of the patients and considered as major in almost 20%.[6] Importantly, OAA adherence may be reduced due to these drug-related problems and may be further complicated by the administration regimen, such as a discontinuous dosing regimen or combination with another OAA or injectable chemotherapy.[7 8] In this context, the prevention of iatrogenic effects is crucial to enhance safety. Furthermore, as drug-related problems lead to an over-consumption of healthcare resources, this could lead to an increase in patient management-related costs.

It has been reported that patients show a preference for OAA over injectable chemotherapy, as the former meets their expectations in terms of comfort and autonomy.[9 10] Their perception of efficacy and past treatments experience (including toxicity) may also influence this preference.[2] It has been shown that for patients to successfully manage their OAA treatment, self-care support and multidisciplinary education across healthcare settings are needed.[11] Finally, such medication management contributes to the optimal effectiveness of the treatment and a better quality of life.[12 13]

Structured OAA programmes have been set up in different countries and described in the literature, based on different components and organisation,[14] highlighting the follow-up and monitoring of patients throughout their treatment.[15] A promising effect has been described in the literature for certain adherence enhancing interventions, applied to target one or multiple factors influencing patient behaviour.[16] The management of side effects should be more specifically included in OAA care programmes in order to help patients better identify them and improve their self-care behaviour.[11] Although studies have also assessed the economic impact of pharmacist-led medication management programmes, showing cost savings and cost avoidance,[17 18] none has evaluated the efficiency of such interventions on the whole patient pathway. There is thus still a need for rigorous studies to assess educational and counselling practices supporting adherence to OAA in terms of clinical benefit and efficiency.[19]

We set up a medico-economic study to evaluate, with a high level of evidence, the cost-effectiveness of a multidisciplinary OAA management programme for outpatients with cancer followed in a French University Hospital.

## METHODS AND ANALYSIS
### Study design and settings
A prospective controlled study with individual open-label randomisation was designed to compare two groups of outpatients treated with OAA at risk of developing a drug-related iatrogenic event and benefiting from follow-up consultations by an oncologist or haematologist from the Lyon Sud Hospital of the *Hospices Civils de Lyon*, France. Patients randomised to the intervention group will benefit from the ONCORAL programme, with enhanced multidisciplinary management involving hospital and community health professionals while patients randomised to the control group will be managed according to usual care.

Each eligible participant will be recruited following the consultation for an initiation or change of OAA with a hospital oncologist or haematologist. If the patient is ambulatory, deemed to be at important risk of developing drug-related iatrogenic events with OAA, and meets all the inclusion criteria (table 1), he will be informed orally and with a detailed written notice by the investigator. Randomisation will be set up after obtaining informed consent, in accordance with the principals of good clinical practice.

The allocation of patients to the two study groups will be done via a centralised online randomisation system, in an unbalanced ratio 2:1 between the ONCORAL programme and usual care. Allocation will not be blinded to participants, medical staff, nor the clinical trial staff.

During the course of the study, in the event of a change in treatment and/or complete discontinuation of OAA, the patient will be withdrawn from the study; however, its data will be considered and analysed.

### Intervention
#### Usual care
The patient is usually followed-up by the hospital oncologist or haematologist every month for the first 3 months after OAA initiation and then, if well tolerated, at 3-month intervals. The aim is to ensure the clinical and biological response to the treatment and to monitor tolerance. Changes in prescriptions can occur if needed (eg, decrease in OAA dosage). There is no intervention from the hospital pharmacist, as OAA is provided by the community pharmacist every month. When dispensing the drug, the community pharmacist may give some advice on how to take it, particularly in relation to other treatments taken by the patient.

#### The ONCORAL programme
The ONCORAL multidisciplinary programme has been described elsewhere.[20 21] Monthly interviews are carried out with a hospital pharmacist and/or a nurse after each consultation with the oncologist or haematologist in the hospital or by telephone if no consultation is scheduled. It is carried out for the entire duration of the OAA treatment and will be assessed in the context of the study up to 12 months after beginning of treatment (ie, patient's inclusion).

Briefly, the therapeutic education sessions are personalised, taking into account the specific needs of each patient. They cover issues relating to understanding the medication plan, including all drugs prescribed to

**Table 1** Patient eligibility criteria

| Inclusion criteria | Exclusion criteria |
|---|---|
| Adult over 18 years of age | Treatment with OAA under a clinical trial or compassionate use |
| Diagnosed cancer | Treatment managed at home by a caregiver exclusively |
| Ambulatory status (not hospitalised for the management and treatment of the cancer) | No declared attending physician |
| Initiation or change of OAA (indication in the context of the marketing authorisation) | No or ≥2 declared regular community pharmacies |
| Sufficient autonomy to manage the treatment at home* | Participation in a clinical trial that may modify the costs of care |
| No major psychiatric cognitive disorder likely to interfere with the ONCORAL programme* | Legal protection or institutionalisation |
| Considered by the oncologist to be at risk of drug-related adverse events, or to have three or more of the following risk factors for developing a drug-related adverse event: | |
| ▶ ≥2 OAA prescribed in combination | |
| ▶ ≥2 lines of treatment | |
| ▶ Combination with an injectable chemotherapy protocol | |
| ▶ Discontinuous OAA regimen | |
| ▶ ≥2 associated chronic conditions | |
| ▶ ≥5 associated drugs including OAA | |
| ▶ Creatinine clearance<60 mL/min | |
| ▶ Frailties and psychosocial conditions (isolated patient, foreigner, and limited autonomy) | |

*According to the treating oncologist.
OAA, oral anticancer agent.

the patient, and the management of side effects and the prevention of drug–drug interaction, including self-medication. If the pharmacist detects an interaction, the prescribing physician is contacted to manage it jointly.

All the community health professionals involved in the patient's care (ie, the attending general practitioner, the community pharmacist, and the home nurse, if any) are informed of the patient's participation in the ONCORAL programme and of all relevant information gathered during the sessions, and are encouraged to share all advice given to the patient using an individual liaison booklet.

### Outcome measures and data collection

The main objective of this medico-economic study is to assess the cost-effectiveness, in terms of cost per quality adjusted life years (QALY), of the ONCORAL programme for the drug management monitoring of outpatients treated with OAA and at risk of developing drug-related iatrogenic events compared with usual care.

#### Primary outcome

The primary outcome will be the estimation of the incremental cost-effectiveness ratio (ICER) at 12 months between the ONCORAL programme and usual care.

The impact of OAA management monitoring will be measured using QALYs that takes into account both survival and quality of life, which is particularly relevant in cancer. The number of QALYs will be assessed using survival time and the 3-level version of the EuroQol 5 Dimensions questionnaire (EQ-5D-3L).[22]

Cost calculations will include hospital-related and ambulatory costs over the 12-month period of the study, from a French health insurance perspective and a hospital perspective.

#### Secondary outcomes

1. Evaluation of OAA management consequences:
   a. Relative-dose intensity (RDI), defined as the ratio of the prescribed dose of the OAA to the dose recommended in the Summary of Product Characteristics approved as part of the marketing authorisation, with a description of the types and reasons for change in dosage.
   b. Adherence to OAA treatment, measured using a 6-item scale combined with the prescription refill rate.
   c. Number and type of adverse drug events related to OAA and their grade (2, 3, or 4) according to the Common Terminology Criteria for Adverse events (CTCAE V.5.0) developed by the US National Cancer Institute.
   d. Number and type of drug–drug interactions related to OAA.

e. Number and type of proven medication errors, with description of their clinical consequences, according to the National Coordinating Council for Medication Error Reporting and Prevention (NCC MERP).

2. Evaluation of clinical outcomes:
   a. Overall survival and progression-free survival.
   b. Cancer-related quality of life, measured using the EORTC QLQ-C30 questionnaire.

3. Description of patient-reported outcomes (PROs) in relation to their treatment:
   a. Patient satisfaction with OAA treatment, measured by the Treatment Satisfaction with Medicines Questionnaire (SATMED-Q).[23]
   b. Patient cognitive and emotional representations of illness, measured by the Brief Illness Perception Questionnaire (Brief IPQ).[24]
   c. Patient cognitive representations of medication, measured by the Beliefs about Medicines Questionnaire (BMQ).[25]
   d. Social support perceived by the patient.[26]
   e. Patient health locus of control according to the Therapeutic Self-Care measure.[27]

4. Budget impact analysis estimating the annual financial impact of the adoption of the ONCORAL programme on the budget of the French national health insurance system.

5. Description of patient and health professional satisfaction with the ONCORAL programme, measured using a visual analogue scale.

### Data collection

The course of the study and the details regarding measurement times for relevant data/questionnaires are presented in online supplemental table. Patients will be managed according to standard practice for the follow-up of their cancer: seven visits will be considered for data collection over the 12-month study period, corresponding to the usual consultations with the oncologist or haematologist at the hospital. At each visit, data will be collected from the hospital clinical records, from a dedicated individual booklet filled in by the patient over time, and from the questionnaires. They will be recorded in an electronic case report form. Onsite data monitoring will be planned, and additional data quality control will be performed throughout the study.

The EQ-5D-3L will be self-administered at baseline and at each follow-up visit. This validated instrument contains five dimensions: mobility, self-care, usual activities, pain/discomfort, and anxiety/depression. Each dimension has three levels: no problems, some problems, and extreme problems.

The following resource use will be collected for costs estimation: medical consultations and laboratory and imaging examinations, performed on an outpatient basis and during hospital stay/consultations; nursing care at home; time off work; drugs taken on an outpatient basis; emergency room admissions; and hospital admissions related to OAA complications. The hospital information system of the participating centre will be used to complete hospitalisation data, as well as the French National Health Service data warehouse (ERASME) for ambulatory data.

A specific microcosting study will be carried out to estimate costs related to the implementation of the ONCORAL programme. Hence, the real time spent by the additional hospital staff required for the intervention will be counted, according to the professional category (ie, pharmacist and nurse). This will be self-reported and will relate to the preparation, realisation, and synthesis for each session; the number of actions carried out towards the community health professionals; and the different additional contacts that could take place with the patient.

### Statistical considerations
#### Sample size calculation

The sample size was based on RDI, as it is associated with the effectiveness of OAA. According to the information available in the literature at the time of the study design, its value was 0.70 in usual care.[28] We assume that the ONCORAL programme will achieve the RDI of 0.85. For an unbalanced 2:1 ratio in favour of the ONCORAL programme, a sample of 130 patients in the ONCORAL group and 65 in the usual care group will achieve a power of 80% at a 0.05 two-sided significance to detect a difference of 0.15 with a SD of 0.35. To take into account early withdrawal from the study, a total of 215 patients will be included in the study (143 in the ONCORAL group and 72 in the usual care group).

The number of patients treated with OAA in the study centre is approximately 350 per year, which will allow the recruitment of the expected number of subjects.

#### Calculation of costs

The calculation of costs will be performed over a 12-month time horizon, which does not require discounting.

From the hospital perspective, the calculation will be based on production costs (ie, gross wages plus employer's costs and social fees) for the time spent by the staff involved in the ONCORAL programme, cost accounting data at hospital level (consultations, examinations, and emergency admissions), and the French national cost studies for hospital stays.

From the French health insurance perspective, the costing will be based on tariffs, that is, the reimbursement rates for ambulatory resources and diagnosis-related groups to estimate the cost for hospital stays. The time spent by the community pharmacists to manage patients included in the ONCORAL programme will be estimated on a sample basis and valued according to expert opinion.

#### Main analyses

The ICER will be expressed as the extra cost in euros per QALY gained following the introduction of the ONCORAL programme compared with the control group:[29]

$$ICER = \frac{C_{Oncoral} - C_{Control}}{QALY_{Oncoral} - QALY_{Control}}$$

where $C_{Oncoral}$ and $C_{Control}$ are the mean cost per patient in each group, and $QALY_{Oncoral}$ and $QALY_{Control}$ the mean number of QALY in each group.

The principle of calculating QALYs is to weight the time spent in health states by the preference scores associated with these states. Multiattribute health state classification systems are one of the methods of valuing health states. The EQ-5D-3L self-administered questionnaire is a multiattribute health state classification system whose preference scores are validated in France.[30] Once the health states are described by the EQ-5D-3L, each of them is individually associated with the corresponding preference score.

A univariate sensitivity analysis will be performed to test the robustness of the cost-effectiveness analysis results and to identify the most sensitive parameters, using a Tornado diagram. The bootstrap method will be used to analyse the sampling uncertainty: 95% CIs of the parameters will be computed; and the joint uncertainty of costs and QALYs will be analysed using cost-effectiveness acceptability curves, which will represent the probability that the ONCORAL programme will be cost-effective at different willingness-to-pay thresholds.

Concerning the primary outcome, a complete case analysis, based on patients for whom all cost and effectiveness data are available, will be performed; an intention-to-treat analysis will be performed in the whole population after imputation of missing data.

Data regarding secondary outcomes will be described for each group. Missing data will be documented but there will be no imputation for secondary outcomes. A Kaplan-Meier model will be used to estimate the overall and the progression-free survival rates up to 12 months in each group, and a log-rank test will be implemented to compare them. Mixed models for repeated measures will be used to estimate the evolution of the different PROs and of cancer-related quality of life.

A pre-established statistical analysis plan will be provided prior to freezing the database and prior to conducting the analyses. All persons involved in data-management, statistical, and medico-economic analysis, as well as the principal investigator and the methodologist will have access to the final data set.

## ETHICS AND DISSEMINATION

The study is funded by a public grant from the French Ministry of Health and sponsored by the *Hospices Civils de Lyon*, a teaching hospital, as responsible for its management. The protocol has been approved by the Ethics Committee on 2 October 2019 (*Comité de Protection des Personnes Ile-de-France VI;* number 59–19), and by the French national data protection agency (*Commission nationale de l'informatique et des libertés*) in March 2020. Its drafting followed the international recommendations

including the Standard Protocol Items: Recommendations for Interventional Trials (SPIRIT) guidelines.[31] All amendments to the protocol will be submitted for approval and communicated (the current protocol version is version 3 of 26 March 2021). Due to the pandemic episode related to COVID-19, inclusions in the study started in December 2020. To date, 191 patients have been included and randomised. The completion date of the study is expected by December 2024.

All eligible patients are informed and given a written notice explaining the objectives of the study and the terms of their participation; they must consent to their participation before being included. Data management complies with the European general data protection regulation. Data are pseudonymised using a unique identification number for each participant; only authorised individuals can access the patients' health information. As no risks related to patient participation in the study are expected, there will be no need to set up an independent monitoring committee.

A scientific committee has been set up to validate the final version of the protocol, to supervise the implementation and conduct of the study, and to draft the resulting reports and publications. Trial results will be disseminated at clinical conferences and published in peer-reviewed journals; the guidelines of the International Committee of Medical Journal Editors (ICMJE) will be followed.

## DISCUSSION

To the best of our knowledge, this medico-economic study is the first randomised controlled trial to evaluate the cost-effectiveness, using QALYs as an effectiveness outcome measure, of a multidisciplinary medication management programme for outpatients treated with OAA.

A recent review retrieved from the literature that most of the OAA adherence programmes were pharmacy or clinic-based, and dedicated to the initiation and/or implementation phase of OAA therapy.[14] The ONCORAL programme is based on the components known to influence adherence to OAA, such as education, counselling, monitoring of adverse events, and drug–drug interactions, adherence follow-up, and a dedicated staff member to be contacted.[32] All sessions are scheduled following a medical consultation with the hospital oncologist, so that patients do not have to return specifically to the hospital, thus reducing the burden. Its originality lies in the involvement of a multidisciplinary hospital team (oncologists, haematologists, pharmacists, and nurses) and the relationship established between them and the community health professionals involved in the follow-up of patients.

Studies reporting pharmacist-led collaborative medication management programmes for OAA have described outcomes to support their benefit, such as patient safety in terms of iatrogenic events, notably drug-related problems that can be identified and solved, and improvement in adherence.[17 33–35] The present study will assess

consequences of OAA management in terms of clinical outcomes: the RDI reflects the changes in prescription that are made as a result of chemotherapy-induced iatrogenic events, and which therefore have an impact on therapeutic effectiveness.[36] The ONCORAL programme should ensure a higher RDI compared with usual care by maintaining adherence to the treatment, notably by educating patients on how to anticipate and manage adverse events. QALYs, which are the remaining years of life for patients and are weighted by a quality of life score, are a particularly important outcome when dealing with cancer patients. QALYs will be used to estimate the cost-effectiveness ratio of the ONCORAL programme compared with usual care, in accordance with the methodological standards of medico-economic studies.[37] To complete the cost analysis of the ONCORAL programme, a microcosting study will be carried out from the hospital perspective, since it is the hospital that is currently financing the additional human resources required for this intervention. This analysis will provide additional and relevant information for hospital decision-makers in order to maintain funding, but also for the French insurance system if payment of a dedicated sum could be considered to extend this intervention to the national level.

Another strength of the study is its pragmatic approach, as the ONCORAL programme is already implemented in the participating centre. There will be no additional intervention for the control group that could impact usual care; patients will only be asked to complete the booklet and questionnaires. Despite the deployment of the ONCORAL programme, not all eligible patients can benefit from it today on a routine basis for financial and logistical reasons. A 2:1 ratio was adopted to randomise patients, as this approximates the number of patients currently being monitored by ONCORAL in current practice.

At the time the study was designed, and to the best of our knowledge, the ONCORAL programme was the only one in France to combine multidisciplinary intervention aimed at all cancer patients with OAA, with a link to community healthcare professionals. Despite the monocentric design of the study, we believe that, if the results are in line with our hypothesis, the ONCORAL programme could be disseminated to centres wishing to implement this intervention. A first step in this direction has already been taken with the introduction of a national experimental care pathway to support French patients taking oral therapies at home.

Some limitations can be outlined. First, blinding was not possible, due to the nature of the intervention. It was also not possible to have a blinded primary outcome assessment as its measure is patient reported; QALYs however is known to be a robust measure. Second, individual rather than period randomisation could be a methodological limitation. However, we observed that the probability that two patients have the same reference community pharmacy is very low, and patients in the control group are not followed-up at the hospital pharmacy. There is therefore no risk of contamination between groups. Finally, the estimation of the number of patients could not be based on the cost-effectiveness outcome, as the assumption on costs and QALYs could not be found in the literature nor from previous data. We therefore made the estimate on the basis of the most relevant clinical outcome, and we believe that this will lead to a sufficient number of patients being included to ensure validation of the ICER analysis.

In conclusion, this study will provide new information on the cost-effectiveness of a specific OAA management programme, ONCORAL. In terms of public health, the results of this study could be used as a model and to share experience, especially as it is a simple and transposable programme. Such an evaluation should also help to promote the securing of OAA management by structuring the city-hospital link, which is all the more important as the obstacles to the development of this type of programme are now well documented, such as the cumbersome nature of implementation, the human resources required, and the lack of funding.[38]

**Author affiliations**
[1]Hospices Civils de Lyon, Pôle de Santé Publique, Service Evaluation Economique en Santé, Lyon, France
[2]Université Lyon 1, Inserm U1290 Research on Healthcare Performance (RESHAPE), Lyon, France
[3]Université Lyon 1, Health Systemic Process, EA 4129 Research Unit, Lyon, France
[4]Hospices Civils de Lyon, Hôpital Lyon Sud, Unité de Pharmacie Clinique Oncologique, Pierre-Bénite, France
[5]Université Lyon 1, CICLY Centre pour l'Innovation et la Cancérologie de Lyon 1-EA3738, Lyon, France

**Acknowledgements** We thank Véréna Landel, from the Direction de la Recherche en Santé of the Hospices Civils de Lyon for her help in manuscript preparation.

**Contributors** LH, PG, GD, and CR designed the study; LH, PG, and GD proposed the methodology and the medico-economic component; AGC, CH, and FR participated in the development of the study. LH, PG, and CR drafted the manuscript; GD, AGC, CH, and FR participated in drafting of the manuscript. All authors have read and approved the final manuscript.

**Funding** The French Ministry of Health (DGOS) supported this work (PRME-17-0487). The funding body had no role in the design of the study.

**Competing interests** None declared.

**Patient and public involvement** Patients and/or the public were not involved in the design, or conduct, or reporting, or dissemination plans of this research.

**Patient consent for publication** Not required.

**Ethics approval** We have deleted the ethical approval statement and data availabilty statement details from the end matter as per style. Please check.

**Provenance and peer review** Not commissioned; externally peer reviewed.

purpose, provided the original work is properly cited, a link to the licence is given, and indication of whether changes were made. See: https://creativecommons.org/licenses/by/4.0/.

**ORCID iD**
Laure Huot http://orcid.org/0000-0002-7870-9912

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
