## [Reviewer comments · BMJ Open]

ARTICLE DETAILS

TITLE (PROVISIONAL)	Cost-effectiveness of the ONCORAL multidisciplinary programme for the management of outpatients taking oral anticancer agents at risk of drug related event: protocol for a pragmatic randomised controlled study
AUTHORS	Huot, Laure; Guerre, Pascale; DESCOTES, Guillaume; CAFFIN, Anne-Gaëlle; HERLEDAN, Chloé; Ranchon, Florence; Rioufol, Catherine

VERSION 1 – REVIEW

REVIEWER	Borget, Isabelle Gustave Roussy
REVIEW RETURNED	28-Aug-2023

GENERAL COMMENTS	Very well written protocol of the cost-effectiveness analysis planned for the ONCORAL intervention. It is well written, the methodology is adequate, clearly described. I have 2 minor comments that authors should add in the discussion section : - the author should discuss the character monocentric of the evaluation. How these results will be extrapolated / applicable to other centre in France after that,- in the same way, it is not clear how the specific cost of the implementation the ONCORAL intervention is funded, what it is perspective of this (is it included in the hospital perspective, in the health system perspective or both ?). Are the ressources consumed included similar between the 2 perspectives ?
--

REVIEWER	Di Nitto, Marco University of Genoa, Health Sciences
REVIEW RETURNED	05-Nov-2023

GENERAL COMMENTS	Thank you for the opportunity to revise this protocol. This study has the potential to be able to report very important information for improving adherence and self-care of patients on oral anticancer agents. The reading of the protocol is fluent and the English is adequate. Some suggestions for improving the manuscript are given below. Abstract The abstract contains all the information necessary to understand the objective of the study and its development Introduction The introduction is complete and it focuses on the main problem of OAA's (i.e., adherence, self-care) that justifies the need for a
--

	program that can enhance this outcome in a cost-effectiveness way. Methods page 9, lines 1-28: In this section, the duration of the intervention should be reported page 12, lines 8-19: This is a very interesting point, as the moment of care is a critical issue, also considering the shortage of healthcare providers. However, it is not entirely clear how it will be measured. Will there be a person in charge of measuring the time of care? Will there be self-reported measurement? Will there be an automatic way to measure it? Page 14, lines 10-19: Will missing data analysis and missing data imputation be performed also for secondary outcomes? Page 14, lines 33-40: The role of the sponsor should be further detailed. The authors reported on page 11, lines 34-36 that the sponsor will monitor the study made by the clinical research associates. Will the sponsor have a role in data collection, data analysis or results reporting? A clear statement regarding the role of the sponsor should be reported. Discussion The discussion section is smooth and the limitations are adequately reported.
--	--

VERSION 1 – AUTHOR RESPONSE

Rewiewer's Comments

• Reviewer 1

Very well written protocol of the cost-effectiveness analysis planned for the ONCORAL intervention. It is well written, the methodology is adequate, clearly described.

I have 2 minor comments that authors should add in the discussion section:

- the author should discuss the character monocentric of the evaluation. How these results will be extrapolated / applicable to other centre in France after that,

Authors' response:

We thank the reviewer for his comment. At the time the study was designed and to our knowledge, the ONCORAL programme was the only one in France to combine pluri-professional intervention that addressed all cancer patients with OAA, with a link to community health professionals. A monocentric design was chosen because the programme was already structured and operational in the investigational centre. If the results of our study are in line with our hypotheses, we believe that the ONCORAL programme could be disseminated in centres wishing to implement this intervention. A first step in this direction has already been taken with the introduction of a national experimental care pathway to support French patients taking oral therapies at home. A paragraph was added in the discussion section to address this issue.

- in the same way, it is not clear how the specific cost of the implementation the ONCORAL intervention is funded, what it is perspective of this (is it included in the hospital perspective, in the health system perspective or both ?). Are the ressources consumed included similar between the 2 perspectives ?

Authors' response:

At present, there is no health insurance cover for the additional costs associated with this intervention. The hospital is therefore willing to fund these human resources costs (pharmacist and nurse). A micro-costing analysis will be performed in the study to measure production costs of the ONCORAL programme, from the hospital perspective. Once these costs are known, provided that the effectiveness of the programme is confirmed, payment of a dedicated sum by the French Health insurance scheme could be considered and estimated on this basis. As suggested by the reviewer, we have added a paragraph in the discussion section to address this issue.

• Reviewer: 2

Thank you for the opportunity to revise this protocol. This study has the potential to be able to report very important information for improving adherence and self-care of patients on oral anticancer agents. The reading of the protocol is fluent and the English is adequate. Some suggestions for improving the manuscript are given below.

Abstract

The abstract contains all the information necessary to understand the objective of the study and its development

Authors' response:

We thank the reviewer for his kind comment.

Introduction

The introduction is complete and it focuses on the main problem of OAAs (i.e., adherence, self-care) that justifies the need for a program that can enhance this outcome in a cost-effectiveness way.

Authors' response:

We thank the reviewer for his kind comment.

Methods

page 9, lines 1-28: In this section, the duration of the intervention should be reported

Authors' response:

The ONCORAL programme is carried out for the entire duration of the OAA treatment. In the context of the study, this means that the intervention takes place up to 12 months after patient's inclusion = beginning of treatment. According to the reviewer advice, we have added this information in the method section.

page 12, lines 8-19: This is a very interesting point, as the moment of care is a critical issue, also considering the shortage of healthcare providers. However, it is not entirely clear how it will be measured. Will there be a person in charge of measuring the time of care? Will there be self-reported measurement? Will there be an automatic way to measure it?

Authors' response:

The time of care will be self-reported by the ONCORAL staff (pharmacist and nurse) at each visit, in the case report form. Precision has been added in the method section.

Page 14, lines 10-19: Will missing data analysis and missing data imputation be performed also for secondary outcomes?

Authors' response:

We choose not to use imputation method in the case of missing data for secondary endpoints. The number of missing data will be presented for each secondary outcome in the results, in order to document this item. We have added this information accordingly.

Page 14, lines 33-40: The role of the sponsor should be further detailed. The authors reported on page 11, lines 34-36 that the sponsor will monitor the study made by the clinical research associates. Will the sponsor have a role in data collection, data analysis or results reporting? A clear statement regarding the role of the sponsor should be reported.

Authors' response:

The study is funded by a public grant from the French Ministry of Health. Funder have no role in the collection, analysis and interpretation of the data, as stated in the manuscript. The Hospices Civils de Lyon is an academic organisation: it is the sponsor as it takes responsibility for the management of the study; but it is its teams (medical and pharmaceutical investigators, statisticians and methodologists) who are involved in the design and conduct of the study. To avoid any misunderstanding, we have amended the sentence on page 11 and added a clarification on funding and sponsor liability in the ethics and dissemination section.

Discussion

The discussion section is smooth and the limitations are adequately reported.

Authors' response:

We thank the reviewer for his kind comment.

VERSION 2 – REVIEW

REVIEWER	Di Nitto, Marco University of Genoa, Health Sciences
REVIEW RETURNED	29-Dec-2023
GENERAL COMMENTS	The authors revised the manuscript following previous comments. There are no further comments.